# Relationship between the Mediterranean Diet and Vascular Function in Subjects with and without Increased Insulin Resistance

**DOI:** 10.3390/nu16183106

**Published:** 2024-09-14

**Authors:** Marta Gómez-Sánchez, Leticia Gómez-Sánchez, Rocío Llamas-Ramos, Emiliano Rodríguez-Sánchez, Luis García-Ortiz, Ruth Martí-Lluch, María Cortés Rodríguez, Inés Llamas-Ramos, Manuel A. Gómez-Marcos

**Affiliations:** 1Home Hospitalization Service, Marqués of Valdecilla University Hospital, s/n, 39008 Santander, Spain; martagmzsnchz@gmail.com; 2Primary Care Research Unit of Salamanca (APISAL), Health Centre of San Juan, Avenida Portugal 83, 2° P, 37005 Salamanca, Spain; leticiagmzsnchz@gmail.com (L.G.-S.); rociollamas@usal.es (R.L.-R.); emiliano@usal.es (E.R.-S.); lgarciao@usal.es (L.G.-O.); inesllamas@usal.es (I.L.-R.); 3Emergency Service, University Hospital of La Paz, Walk. of Castellana, 261, 28046 Madrid, Spain; 4Institute of Biomedical Research of Salamanca (IBSAL), University Hospital of Salamanca, Walk of San Vicente, 182, 37007 Salamanca, Spain; 5Faculty of Nursing and Physiotherapy, Universidad de Salamanca, 37007 Salamanca, Spain; 6Red de Investigación en Cronicidad, Atención Primaria y Promoción de la Salud (RICAPPS), 37005 Salamanca, Spain; rmarti.girona.ics@gencat.cat; 7Primary Healthcare Management, Castilla y León Regional Health Authority (SACyL), 37007 Salamanca, Spain; 8Department of Medicine, University of Salamanca, 37007 Salamanca, Spain; 9Department of Biomedical and Diagnostic Sciences, University of Salamanca, 37007 Salamanca, Spain; 10Vascular Health Research Group, Instituto Universitario para la Investigación en Atención Primaria de Salud Jordi Gol i Gurina (IDIAPJGol), 08007 Girona, Spain; 11Girona Biomedical Research Institute (IDIBGI), Doctor Trueta University Hospital, 17190 Girona, Spain; 12Department of Medical Science, Faculty of Medicine, University of Girona (UdG), 17003 Girona, Spain; 13Department of Statistics, University of Salamanca, 37008 Salamanca, Spain; 14Department of Hematology, University Hospital of Salamanca, 37008 Salamanca, Spain

**Keywords:** vascular function, insulin resistance, Mediterranean diet

## Abstract

(1) The main aim of this study was to analyze the relationship of the Mediterranean diet (MD) with vascular function in participants with and without increased insulin resistance (IR) in the Spanish population. A secondary aim was to study differences by gender. (2) Methods: Data were analyzed from 3401 subjects in the EVA, MARK, and EVIDENT studies (mean age = 60 years and 57% men). IR was evaluated with the triglyceride and glucose index (TyG index). TyG index = Ln [(fasting triglyceride mg/dL × fasting glucose mg/dL)/2]. The MD was measured against the MEDAS questionnaire, with the 14 items used in the PREDIMED study. Vascular stiffness was estimated with the brachial–ankle pulse wave velocity (baPWV) and the cardio ankle vascular index (CAVI) using the Vasera VS-1500^®^. (3) Results: The mean MEDAS value was 5.82 ± 2.03; (men: 5.66 ± 2.06; women: 6.04 ± 1.99; *p* < 0.001). MD adherence was 36.8% (men: 34.2%; women: 40.3%; *p* < 0.001). The mean baPWV value was 14.39 ± 2.78; (men: 14.50 ± 2.65; women: 14.25 ± 2.93; *p* = 0.005). A baPWV value ≥ 14.5 m/s was found in 43.4% (men: 43.6%; women: 40.0%; *p* = 0.727). The mean CAVI value was 8.59 ± 1.28; (men: 8.75 ± 1.28; women: 8.37 ± 1.26; *p* < 0.001). CAVI values ≥ 9 were present in 39.0% (men: 44.4%; women: 31.7%; *p* < 0.001). The mean value of the TGC/G index was 10.93 ± 1.39; (men: 11.08 ± 1.33; women: 10.73 ± 1.43; *p* < 0.001). IR was found in 49.9%. The average value of the MD score value was negatively associated with baPWV and CAVI in all groups analyzed (<0.05), except in the group of women with insulin resistance. (4) Conclusions: The results suggest that MD adherence is negatively associated with the vascular stiffness parameters analyzed in all the groups studied except the group of women with insulin resistance.

## 1. Introduction

The Mediterranean diet (MD) is a widely studied dietary pattern and has been associated with numerous health benefits. The MD is recognized as a good evidence-based model for the prevention of cardiovascular disease (CD) [1,2,3,4,5]. Two major randomized trials in secondary prevention (the Lyon Heart study [6] and the CORDIOPREV study [7]) and one primary prevention trial (the PREDIMED study [8]) have demonstrated these benefits. Also, there are several prospective studies that support the health benefits of the MD [9,10,11,12]. The ATTICA study showed that each increase of 1 unit in the baseline MD score was associated with an 8% decrease in the incidence of CD at 20 years of age [13]. The MD is characterized by a higher intake of extra virgin olive oil, natural vegetables (fruits, vegetables, nuts, and legumes), and fish, accompanied by a lower intake of processed meats, red meats, and ultra-processed products, and a moderate consumption of wine [1,2,3,4,5]. It has become one of the healthiest dietary patterns, integrating numerous elements with antioxidant and anti-inflammatory properties [12,14,15,16], since its composition of micronutrients and macronutrients is suitable for health [17]. Data published in several studies, including reviews and meta-analyses, have demonstrated the beneficial effects of the MD on chronic diseases, such as type 2 diabetes mellitus [10,18], obesity [19], cardiovascular diseases [20], cancer [19], and general morbidity and mortality [9,12,15]. Thus, the MD is currently considered a useful tool to prevent cardiovascular diseases and other health problems [4,12,15].

Arterial stiffness (AS) decreases the elasticity of the arteries and is a risk predictor for CD of similar or greater importance than other traditional cardiovascular risk factors [21]. Measuring AS by oscillometry in a noninvasive manner, both with the brachial–ankle pulse wave velocity (baPWV) [22] and the cardio–ankle vascular index (CAVI) [23,24], provides useful measures in practice to assess peripheral and central arterial stiffness. Both measures have shown a positive association with cardiovascular events [25,26,27,28], reflect the dissociation between the chronological and biological ages of large arteries, and are determined primarily by age, sex, and blood pressure [21,29]. As the central arteries become stiffer, they maintain their conduit function but progressively lose their storage property [30]. AS is linked to cardiovascular risk factors, lifestyle, inflammatory factors, and psychological factors [31,32,33,34,35]. The detection of AS can, therefore, play an important role in the prevention of cardiovascular diseases since it appears before the vascular structure is affected [30]. Furthermore, the MD is associated with reduced inflammation, improved endothelial function, and AS [36,37,38]. Meanwhile, some studies have shown a relationship between AS measured with baPWV and insulin resistance (IR) [39,40].

Insulin resistance (IR), a decreased response to insulin in target tissues, causing hyperinsulinemia, is a pathological condition that underlies different cardiovascular risk factors, such as type 2 diabetes mellitus [41], abdominal obesity [42], hypertension [43], cardiovascular diseases [43], metabolic syndrome [44], cancer [45], and depression, especially in the elderly [46,47]. Therefore, measuring IR can be of considerable help in clinical practice, and using therapeutic measures to treat IR may be a potential strategy to prevent CD [48,49]. It is also known that women are more sensitive to insulin compared to men, but this metabolic advantage gradually disappears after menopause or when IR progresses to hyperglycemia and diabetes [50]. However, the gold standard for calculating IR, namely the glucose clamp technique, is an expensive and complicated procedure. This makes it difficult in clinic, and it is, therefore, used only in experimental studies [48]. For this reason, different parameters have been generated and used that measure IR indirectly [49,51]. Among these is the triglyceride/glucose index (TyG index), which has shown an association with AS and coronary artery disease in different studies [52,53], stroke risk in Chinese subjects [54], and cardiovascular morbidity and mortality [55,56], making it a potentially valuable tool for understanding and predicting the risk of cardiovascular disease. Similarly, IR has been associated with increased AS [40,57,58].

However, studies analyzing the relationship of the Mediterranean diet with two parameters that measure peripheral and central vascular function in participants with and without increased IR are scarce. We, therefore, propose this study, with its main aim being to analyze the relationship of the Mediterranean diet with vascular function in participants with and without increased IR in the Spanish population, and a secondary aim to study the differences found between genders.

## 2. Methods

### 2.1. Study

This study includes the variables of the participants included in the EVA study [59] (NCT02623894), the variables of the participants included in the MARK study [60] (NCT01428934), and the variables of the participants included in the EVIDENT study [61] (NCT0108308). All subjects from these studies were included whose data records for the key variables analyzed in the present study were complete. The main objective of the EVA study [59] was to analyze the influence of cardiovascular risk factors, lifestyles, and psychological and inflammatory factors on vascular aging, in an urban population aged between 45 and 74 without cardiovascular disease at the time of inclusion. The main objective of the MARK study [60] was to analyze whether the brachial–ankle index, arterial stiffness, postprandial glucose, HbA1c, and blood pressure self-measured by the patient at home could improve the predictive capacity of the classical equations to estimate cardiovascular risk in a population aged between 35 and 74 years with intermediate cardiovascular risk. The main objective of the EVIDENT study [61] was to develop and validate a smartphone application, and to measure the effect of this tool together with a standardized intervention designed to improve the MD and physical activity. The results presented were the secondary objectives of the studies analyzed.

### 2.2. Population

These studies were carried out in primary care. A total of 3401 people were included (men: 1943; women: 1458), aged between 35 and 75 years. Specifically, 490 of the 501 participants in the EVA study were included [59]. In the EVA study, the subjects were selected by random sampling of the urban population without CD (reference population: 43,946). Of the 2511 subjects recruited in the MARK study, subjects with intermediate cardiovascular risk were randomly sampled from seven urban health centers, and 2468 were included [60]. The EVIDENT study [61] had 1104 subjects selected by random sampling from a primary care center, of which 443 participants were included here. Figure 1 shows a flowchart of this study. It shows the number of subjects included, as well as the number of subjects excluded from each of the studies analyzed in this work.

### 2.3. Committee Ethical and Consent to Participate

The protocols of the three studies were approved by the Ethics Committee for Research with Medicines of Salamanca. The registration numbers assigned were PI15/01039 and PI20/10569 (EVA study [59]), approval date: 4 May 2015), PI10/02043 (MARK study [60], approval date: 3 December 2013), and PI83/06/2018, (EVIDENT study [61]), approval date: 25 April 2016). The subjects signed a consent form before being included. All standards established in the Declaration of Helsinki and the quality standards established by the WHO for observational studies were followed during the study [62].

### 2.4. Variables and Measuring Instruments

The collection of each and every one of the variables is described in the protocols of the different studies previously published [59,60,61].

#### 2.4.1. Mediterranean Diet

The MD was registered following the recommendations indicated in the PREDIMED study questionnaire [63]. The questionnaire has previously been validated in Spanish adults [63]. The questionnaire comprises 14 items, of which 12 include questions about the frequency of food consumption, and 2 others include questions about the typical eating habits of the Spanish population. Each question is scored with 0 or 1. The following are assessed as 1 point: the fat for cooking is olive oil; consumption of olive oil ≥ to 4 tablespoons; consumption of vegetables ≥ of 2 portions; consumption of fruit ≥ 3 pieces; consumption of red meat or processed meat < 1 serving; consumption of animal fat < 1; consumption of sugary drinks < 100 mL; and intake of more white meat than red meat. All of these refer to daily consumption. Per week, the following are assessed as 1 point: consumption of wine ≥ 7 glasses; legumes ≥ 3 portions; fish ≥ 3 portions; nuts or nuts ≥ 3; stir-fry ≥ 2 portions; and < 2 baked goods. The value of this questionnaire ranges from 0 to 14 points. We considered good adherence to the MD when the score was > the median value [63].

#### 2.4.2. Insulin Resistance

To assess insulin resistance, we used the triglyceride and glucose index (TyG index), estimated with the following formula: TyG index = Ln [(fasting triglyceride in mg/dL × fasting glucose in mg/dL)/2] [64]. The median value in men and women was used as a cut-off point to classify subjects with insulin resistance.

#### 2.4.3. Arterial Stiffness Measurement

Evaluations of the baPWV and the CAVI were carried out with the VaSera VS-1500 device (Fukuda Denshi Co, Ltd., Tokyo, Japan), following the manufacturer’s instructions [23]. For 10 min prior to the measurement, the patients had to have been at rest. The hour before the measurement, they could not have smoked or ingested caffeine. A total of 4 cuffs were placed on the arms and legs adapted to their circumference. A microphone with adhesive tape on both sides was placed in the second intercostal space. CAVI measurements were considered valid when the device recorded at least 3 consecutive valid beats. The VaSera VS-1500 device estimates the values of the CAVI automatically calculated by substituting the stiffness parameters in the following equation to detect the vascular elasticity stiffness parameter *β* = 2 ρ × 1/(Ps − Pd) × ln (Ps/Pd) × PWV2, where ρ is the blood density, Ps and Pd are the SBP and DBP in mmHg, and the PWV was measured between the aortic valve and the ankle [23]. The baPWV was calculated with the following formula: baPWV = ((0.5934 × height (cm) + 14.4724))/tba. In this equation, tba is the time between the arm and ankle waves [23].

#### 2.4.4. Cardiovascular Risk Factors

All measurements were performed in primary care following previously published recommendations in three protocols [59,60,61]. We considered if patients the met any of the following conditions: hypertensive (they were being treated with antihypertensive drugs or had blood pressure levels ≥ 140/90 mmHg); diabetics (they were on anti-diabetes drugs or had fasting plasma glucose levels ≥ 126 mg/dL or HbA1c values ≥ 6.5%); dyslipidemic (they were on lipid-lowering drugs or had fasting total cholesterol values ≥ 240 mg/dL, low-density lipoprotein cholesterol values ≥ 160 mg/dL, high-density lipoprotein cholesterol values ≤ 40 mg/dL in men and ≤ 50 mg/dL in women, or triglycerides values ≥ 150 mg/dL); obese (if the BMI value ≥ 30 kg/m^2^) [59,60,61].

### 2.5. Statistical Analysis

The normal distribution of the main variables was checked with the Kolgomorov–Smirnov test. Continuous variables are shown as the mean ± SD. The categorical variables are shown as the number and %. Student’s *t* test or the chi-square test were used to estimate the differences between the sexes, as appropriate. The correlation between the continuous variables was performed with the Pearson correlation coefficient.

Different multiple linear regression models were used to analyze the association between the values of CAVI and baPWV with the MD. The mean values of the CAVI and baPWV were independent variables, and the mean value of MD adherence was the dependent variable. In all models, age, gender, and treatment with antihypertensive, hypoglycemic, and lipid-lowering drugs were used as adjustment variables. All analyses were performed overall, by gender, and in subjects with and without IR.

The new variables needed to carry out this work were generated with the SPSS Statistics program for Windows, version 28.0 (IBM Corp, Armonk, NY, USA). We considered statistical significance when the *p* value was <0.05.

## 3. Results

### 3.1. Description of Mediterranean Diet and Other Variables Analyzed Overall and by Gender

This study analyzed the data from 3401 subjects from the EVA, MARK, and EVIDENT studies with a mean age of 60.14 ± 9.77 years (1943 men, 1458 women). The mean MEDAS score was 5.82 ± 2.03; (males: 5.66 ± 2.06; females: 6.04 ± 1.99; *p* value < 0.001). MD adherence was found in 36.8% (males: 34.2%, females: 40.3%; *p* value < 0.001). The mean baPWV score was 14.39 ± 2.78; (males: 14.50 ± 2.65, females: 14.25 ± 2.93; *p* value = 0.005). A baPWV value of ≥ 14.5 m/s was found in 43.4% (males: 43.6%, females: 40.0%; *p* value = 0.727). The mean CAVI value was 8.59 ± 1.28; (males: 8.75 ± 1.28, females: 8.37 ± 1.26; *p* value < 0.001). A CAVI value of ≥9 was found in 39.0% (males: 44.4%, females: 31.7%; *p* value < 0.001). The mean TyG index value was 10.93 ± 1.39; (males: 11.08 ± 1.33; females: 10.73 ± 1.43; *p* value < 0.001). IR was found in 49.9% (no difference between the sexes). Table 1 reflects the general characteristics of the participants.

### 3.2. Mediterranean Diet, Risk Factors, and Vascular Function According to Insulin Resistance

The subjects with IR had lower MD adherence (5.49 ± 1.96 vs. 6.16 ± 2.06), were older, had higher blood pressure, total cholesterol, LDL cholesterol, triglycerides, glycemia, HbA1c, weight, BMI, and WC, and lower HDL cholesterol values than the subjects without insulin resistance, as assessed with the TyG index. The subjects with IR had higher baPWV (14.91 ± 2.70 vs. 13.87 ± 2.75) and CAVI (8.73 ± 1.25 vs. 8.45 ± 1.31) values than those without IR (Table 2).

In the analysis by sex, the men with IR had lower MD adherence (5.32 ± 2.01 vs. 6.00 ± 2.05), were younger, had higher blood pressure, total cholesterol, LDL cholesterol, triglycerides, glycemia, HbA1c, weight, BMI, and WC, and lower HDL cholesterol values than the men without insulin resistance, as assessed with the TyG index. The men with IR showed lower baPWV values with no differences in the CAVI (Table 3). The women with IR showed lower MD adherence (5.72 ± 1.86 vs. 6.36 ± 2.05), were older, had higher blood pressure, total cholesterol, LDL cholesterol, triglycerides, glycemia, HbA1 c, weight, BMI, and WC, and lower HDL cholesterol values than the women without IR, as assessed with the TyG index. The women with IR had higher baPWV (15.18 ± 2.79 vs. 13.32 ± 2.77) and CAVI (8.65 ± 1.20 vs. 8.09 ± 1.25) scores (Table 4).

Figure 2 shows the distribution of the participants analyzed according to whether or not they presented IR and MD adherence.

The differences in the TyG index between the participants with and without cardiovascular risk factors overall and by sex are shown in Figure 3.

### 3.3. Correlation between Mediterranean Diet and Vascular Function Parameters, Overall and by Sex

Table 5 presents the correlation of MD with the different cardiovascular risk factors of vascular function and the TyG index, overall, in subjects with and without IR, and by gender. In the overall analysis, the MD showed a negative correlation with the different cardiovascular risk factors except for age.

### 3.4. Association between Mediterranean Diet and Vascular Function Overall and by Sex in Subjects with and without IR, Assessed with Multiple Regression Analysis

The results of the multiple regression analysis are shown in Table 6. The mean value of the MD score showed a negative association with the baPWV and CAVI in all groups analyzed (<0.05), except in the group of women with insulin resistance; although the association was negative, it did not reach statistical significance (baPWV, *p* = 0.549 and CAVI, *p* = 0.265).

## 4. Discussion

The results of this study carried out on 3401 Spanish subjects show that 37% adhered to the MD, and 40% had increased arterial stiffness. However, these proportions were different between the sexes, with men having lower Mediterranean diet adherence and higher AS and IR values than women. A negative association was found between Mediterranean diet and the AS parameters analyzed, although there were differences between the sexes in the group of subjects with IR, where the association was found in males but not in females.

Similar to the data from other authors, MD adherence was greater in women [65,66]. However, in several studies carried out mainly in younger populations, there were no differences between the sexes [67,68,69]. Therefore, greater MD adherence in women is not clear, and there are likely other more important factors than sex, such as age and socioeconomic or educational levels [68,70,71]. In any case, there is considerable evidence, as reflected in numerous cross-sectional and cohort studies [15,16]. These results have been corroborated in different meta-analyses and systematic reviews, such as Refs. [1,5,68,72], which have demonstrated the benefits of the MD to reduce morbidity and mortality from CD [9,12,18,73], in addition to preventing cardiovascular risk factors, such as diabetes mellitus [10,18], obesity [19], and various types of cancer [16]. Moreover, the MD offers long-term protection against CDs, some of which are mediated by inflammation, uricemia, and renal function [74]. For all these reasons, the MD is a healthy lifestyle that can improve a population’s health [1,12,15]. However, despite the benefits of the MD described in different studies, there are authors who have indicated the existence of biases in the research, such as a lack of a consensus definition of the MD, differences among populations, and other factors that may have influence, such as socioeconomic status and other lifestyles [75,76]. The beneficial effects are probably due to the MD’s high antioxidant and fiber content and lower saturated fat content [5,12,14,15,16].

In this work, we found negative association between the MD adherence score and peripheral AS values measured with baPWV, and central and peripheral AS, as assessed with the CAVI, i.e., a greater use of the MD was associated with lower baPWV and CAVI values. These results are consistent with the data published in several clinical trials, such as the NU-AGE clinical trial [35]. In the study, they found that the Mediterranean-style diet was effective for cardiovascular health and decreased blood pressure and AS. Similarly, the trial endorsed by the American College of Physicians concluded that a healthy dietary intervention, such as the MD, promoted the regression of proximal aortic stiffness, independently of weight loss [77]. However, the EVasCu study found that subjects with greater adherence to the MD showed greater arterial stiffness, overall and in women, although this difference disappeared when adjusted for age [37]. This may be explained by the fact that as people age, their health problems increase and they are more likely to adopt healthier dietary patterns [18]. On the other hand, AS increases with age due to a loss of elasticity and increased fibrosis in the arteries [78]. This may diminish the benefits that the MD can cause [79]. Therefore, an increase in AS in older subjects and greater adherence to the MD could be a confounding variable. In these cases, age can be an intermediate variable, influencing dietary adherence and vascular health. The mechanisms that can explain this association are based on the fact that the abundant fiber, antioxidants, vitamins, and monounsaturated fatty acids provided by the MD are associated with a decrease in inflammation and an improvement in endothelial function, leading to a reduction in AS [36,37,38,80,81,82].

In this study, the subjects with IR, as assessed with the TyG index, had higher baPWV and CAVI values. This is in line with many studies carried out in recent years showing a positive association between the TyG index and AS, including a study carried out on 912 Chinese subjects [39]. The Kailuan study of 6028 participants showed that a 1-unit increase in TyG resulted in a 39 cm/s increase in baPWV (*p* value < 0.001) [40]. Similar results have been gathered in several meta-analyses. One included a total of 9 observational studies, with a total of 37,780 participants, and concluded that a higher TyG ratio was associated with greater probabilities of subclinical atherosclerosis and AS [83]. Another included 13 observational studies with 48,332 participants. Of these, two were prospective cohort studies, and the remaining eleven were cross-sectional studies. It was found that the risk of developing high AS was 1.85 times higher for those in the subgroup with the highest TyG ratio versus the lowest group, concluding that a high TyG ratio was associated with a higher incidence of AS [84]. The TyG ratio can, therefore, be used as an independent predictor of a higher risk of AS, and it can also be considered as a tool in primary care to identify people at high risk of CD, making early intervention possible for different risk factors. However, studies must first be carried out to establish the best cut-off point that would allow professionals to identify subjects with increased insulin resistance.

Finally, in this study, we found that the MD had a negative association with AS in all groups except in women with increased IR. Similarly, some earlier studies have suggested that using the MD only leads to significant improvements in insulin homeostasis in men [85]. However, there are also studies that suggest that women may reduce their cardiovascular risk to a greater extent than men when they show greater adherence to the MD. [4,25]. It is also known that women are more sensitive to insulin compared to men, but this metabolic advantage gradually disappears after menopause or when IR progresses to hyperglycemia and diabetes [50]. Also, there are studies that suggest that the vegetarian diet is more followed by women than by men. This, together with hormonal and metabolic differences, would improve AS [38]. The role of sex in the link between the MD and AS in subjects with higher IR is, therefore, not clear, and prospective studies are required to clarify this relationship.

Limitations and strengths: (a) the analysis of cross-sectional data prevented establishing causality; (b) the results of adults without previous cardiovascular disease prevented their generalization; (c) the findings came from three studies with different characteristics among the number of subjects included from each of them, with the largest number of people included coming from the MARK study; (d) urban origin, not extrapolated to the rural population; (e) the cut-off points used to classify insulin resistance and adherence to the MD were not agreed; therefore, the median value is arbitrary; (f) the measurements were carried out by different researchers; however, all received previous training; (g) finally, the MD data were collected with questionnaires and may include information bias. Nevertheless, this study also has several strengths. It is the first study on a large sample of the Spanish population carried out in primary care that jointly analyzes the relationship between the MD and AS in subjects with and without insulin resistance. Furthermore, all measurements were performed under standardized conditions and with validated devices.

## 5. Conclusions

The results of this work suggest that adherence to the MD has a negative association with the AS parameters analyzed in all the groups assessed except the group of women with insulin resistance.

## Figures and Tables

**Figure 1 nutrients-16-03106-f001:**
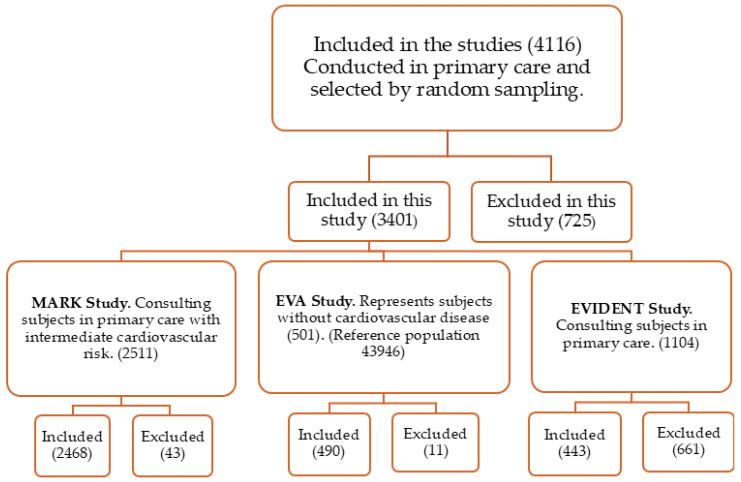
The flowchart shows the number of included and excluded subjects from each of the three studies analyzed in this paper.

**Figure 2 nutrients-16-03106-f002:**
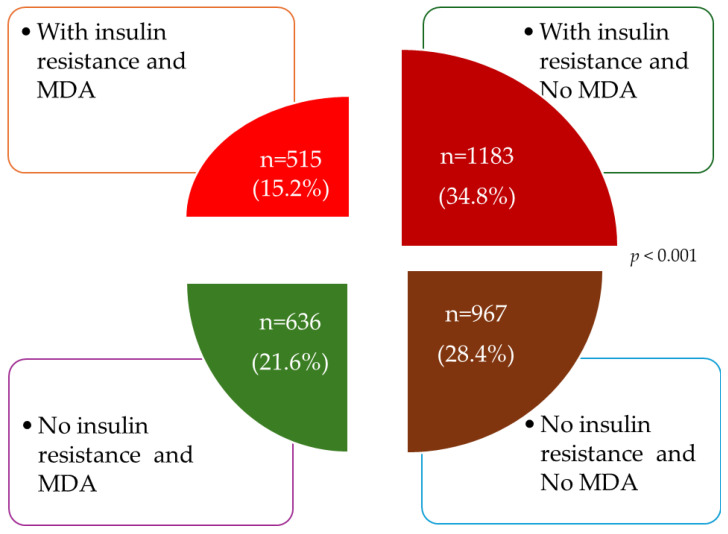
Distribution of participants according to whether they had insulin resistance or MD adherence. MDA: Mediterranean diet adherence. *p*: differences between groups.

**Figure 3 nutrients-16-03106-f003:**
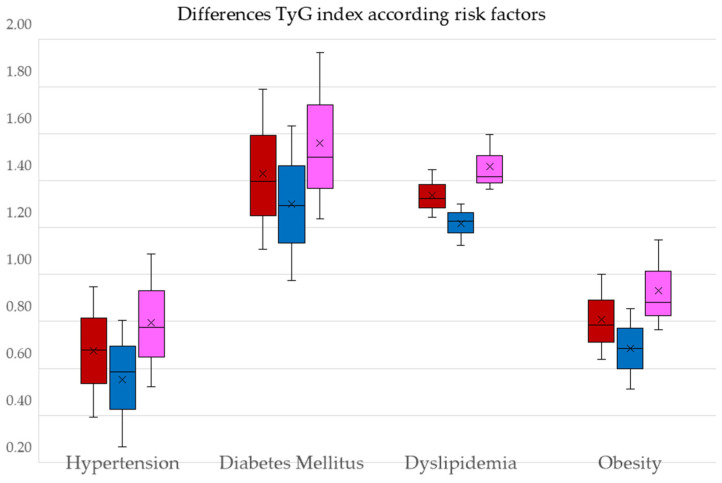
Differences and 95% CI in the mean value of the TyG index between participants with and without. Cardiovascular risk factors. TyG index: triglycerides/glucose index. The red bars represent all participants. The blue bars represent men, and the pink bars represent women.

**Table 1 nutrients-16-03106-t001:** General characteristics of the participants overall and by gender.

	Overall (n = 3401)	Women (n = 1458)	Men (n = 1943)	*p* Value
Mediterranean diet				
MD, total score	5.82 ± 2.03	6.04 ± 1.98	5.66 ± 2.06	<0.001
MD adherence, n (%)	1251 (36.8)	587 (40.3)	664 (34.2)	<0.001
Conventional risk factors				
Age, years	60.14 ± 9.77	60.03 ± 10.21	60.23 ± 9.58	0.559
SBP, mmHg	133.34 ± 19.37	128.71 ± 20.69	136.82 ± 17.55	<0.001
DBP, mmHg	81.95 ± 10.94	79.58 ± 10.82	83.74 ± 10.69	<0.001
Hypertension, n (%)	2180 (64.1)	844 (57.9)	1336 (68.8)	<0.001
Antihypertensive drugs, n (%)	1557 (45.8)	649 (44.5)	908 (46.7)	0.199
Total cholesterol, mg/dL	216.15 ± 41.25	220.08 ± 42.79	213.20 ± 39.81	<0.001
LDL cholesterol, mg/dL	132.91 ± 35.19	132.17 ± 35.95	133.47 ± 34.61	0.286
HDL cholesterol, mg/dL	52.57 ± 14.47	57.28 ± 15.73	49.03 ± 12.31	<0.001
Triglycerides, mg/dL	131.98 ± 75.29	121.59 ± 63.52	139.79 ± 82.19	<0.001
Dyslipidemia, n (%)	2453 (72.1)	1064 (73.0)	1389 (71.5)	0.338
Lipid–lowering drugs, n (%)	965 (28.4)	426 (29.2)	539 (27.7)	0.344
FPG, mg/dL	101.55 ± 31.42	100.18 ± 32.34	102.57 ± 30.68	0.028
HbA1c, %	5.94 ± 1.05	5.93 ± 1.07	5.94 ± 1.03	0.787
Diabetes mellitus, n (%)	751 (22.1)	305 ± 20.9	446 (23.0)	0.157
Hypoglycemic drugs, n (%)	570 (16.8)	234 (16.0)	334 (17.3)	0.337
Weight, kg	77.45 ± 14.62	70.18 ± 13.33	82.91 ± 13.10	<0.001
Height, cm	164.56 ± 9.46	157.10 ± 6.69	170.15 ± 7.07	<0.001
BMI, kg/m^2^	28.54 ± 4.53	28.47 ± 5.30	28.60 ± 3.86	0.407
WC, cm	98.60 ± 12.09	94.31 ± 12.91	101.82 ± 10.33	<0.001
Obesity, n (%)	1072 (31.5)	492 (33.7)	580 (29.9)	0.016
TyG Index	10.93 ± 1.39	10.73 ± 1.13	11.08 ± 1.33	<0.001
TyG Index, n (%)	1698 (49.9)	728 (49.9)	970 (49.9)	0.996
Vascular function				
CAVI	8.59 ± 1.29	8.37 ± 1.26	8.75 ± 1.28	<0.001
CAVI ≥ 9	1325 (39.0)	462 (31.7)	863 (44.4)	<0.001
baPWV, m/s	14.39 ± 2.78	14.25 ± 2.93	14.50 ± 2.65	0.010
baPWV ≥ 14.5, m/s	1475 (43.4)	627 (43.0)	848 (43.6)	0.709

Continuous variables are displayed as means ± SD. Categorical variables are shown as number and %. MD: Mediterranean diet; SBP: systolic blood pressure; DBP: diastolic blood pressure; LDL: low-density lipoprotein; HDL: high-density lipoprotein; FPG: fasting plasma glucose; HbA1c: glycosylated hemoglobin; BMI: body mass index; WC: waist circumference; TyG Index: triglycerides/glucose index; CAVI: cardio–ankle vascular index; baPWV: brachial–ankle pulse wave velocity. *p* value: differences between sexes.

**Table 2 nutrients-16-03106-t002:** General characteristics of the patients with and without IR.

	Insulin Resistance (n = 1698)	No Insulin Resistance (n = 1703)	*p* Value
Mediterranean diet			
MD, total score	5.49 ± 1.96	6.16 ± 2.06	<0.001
MD adherence, n (%)	515 (30.3)	736 (43.2)	<0.001
Conventional risk factors			
Sex (women), n (%)			
Age, years	60.92 ± 8.35	59.37 ± 10.95	<0.001
SBP, mmHg	136.45 ± 17.55	130.25 ± 20.58	<0.001
DBP, mmHg	83.99 ± 10.53	79.93 ± 10.97	<0.001
Hypertension, n (%)	1254 (73.9)	926 (54.4)	<0.001
Antihypertensive drugs, n (%)	921 (54.2)	636 (37.3)	<0.001
Total cholesterol, mg/dL	223.29 ± 42.21	209.02 ± 39.00	<0.001
LDL cholesterol, mg/dL	137.16 ± 36.65	128.72 ± 33.17	<0.001
HDL cholesterol, mg/dL	47.11 ± 10.96	58.01 ± 15.46	<0.001
Triglycerides, mg/dL	178.91 ± 98	85.19 ± 23.62	<0.001
Dyslipidemia, n (%)	1533 (90.3)	920 (54.0)	<0.001
Lipid-lowering drugs, n (%)	586 (34.5)	379 (22.3)	<0.001
FPG, mg/dL	114.40 ± 37.94	88.73 ± 14.39	<0.001
HbA1c, %	6.30 ± 1.26	5.57 ± 0.59	<0.001
Diabetes mellitus, n (%)	592 (34.9)	159 (9.3)	<0.001
Hypoglycemic drugs, n (%)	442 (26.0)	128 (7.5)	<0.001
Weight, kg	80.78 ± 14.87	74.13 ± 13.59	<0.001
Height, cm	164.31 ± 9.64	164.81 ± 9.27	0.122
BMI, kg/m^2^	29.87 ± 4.54	27.23 ± 4.13	<0.001
WC, cm	102.15 ± 11.27	95.06 ± 11.83	<0.001
Obesity, n (%)	705 (41.5)	367 (21.6)	<0.001
TyG Index	12.01 ± 1.03	9.85 ± 0.69	<0.001
Vascular function			
CAVI	8.73 ± 1.25	8.45 ± 1.31	<0.001
CAVI ≥ 9	738 (43.5)	587 (34.5)	<0.001
baPWV, m/s	14.91 ± 2.70	13.87 ± 2.75	<0.001
baPWV ≥ 14.5, m/s	860 (50.6)	615 (36.1)	<0.001

Continuous variables are displayed as means ± SD. Categorical variables are shown as number and %. MD: Mediterranean diet; SBP: systolic blood pressure; DBP: diastolic blood pressure; LDL: low-density lipoprotein; HDL: high-density lipoprotein; FPG: fasting plasma glucose; HbA1c: glycosylated hemoglobin; BMI: body mass index; WC: waist circumference; TGC: triglycerides; TyG Index: triglycerides/glucose index; CAVI: cardio–ankle vascular index; baPWV: brachial–ankle pulse wave velocity. *p* value: differences between subjects with and without insulin resistance.

**Table 3 nutrients-16-03106-t003:** General characteristics of the male patients with and without IR.

	Insulin Resistance (n = 970)	No Insulin Resistance (n = 973)	*p* Value
Mediterranean diet			
MD, total score	5.32 ± 2.01	6.00 ± 2.05	<0.001
MD adherence, n (%)	274 (28.2)	390 (40.1)	<0.001
Conventional risk factors			
Age, years	59.76 ± 8.67	60.70 ± 10.38	0.031
SBP, mmHg	138.38 ± 16.92	135.26 ± 18.02	<0.001
DBP, mmHg	85.53 ± 10.46	81.95 ± 10.63	<0.001
Hypertension, n (%)	722 (74.4)	614 (63.1)	<0.001
Antihypertensive drugs, n (%)	499 (51.4)	409 (42.0)	<0.001
Total cholesterol, mg/dL	219.13 ± 40.77	207.28 ± 37.93	<0.001
LDL cholesterol, mg/dL	135.21 ± 35.73	131.76 ± 33.41	0.029
HDL cholesterol, mg/dL	44.70 ± 10.32	53.35 ± 12.63	<0.001
Triglycerides, mg/dL	190.08 ± 89.03	89.65 ± 23.56	<0.001
Dyslipidemia, n (%)	869 (89.6)	520 (53.4)	<0.001
Lipid-lowering drugs, n (%)	310 (32.0)	229 (23.5)	<0.001
FPG, mg/dL	113.86 ± 37.19	91.32 ± 15.72	<0.001
HbA1c, %	6.24 ± 1.23	5.64 ± 0.65	<0.001
Diabetes mellitus, n (%)	324 (33.4)	122 (12.5)	<0.001
Hypoglycemic drugs, n (%)	239 (24.6)	97 (10.0)	<0.001
Weight, kg	85.74 ± 13.87	80.09 ± 11.62	<0.001
Height, cm	170.26 ± 6.86	170.05 ± 7.28	0.520
BMI, kg/m^2^	29.52 ± 3.95	27.68 ± 3.53	<0.001
WC, cm	104.10 ± 10.44	99.55 ± 9.70	<0.001
Obesity, n (%)	361 (37.2)	219 (22.5)	<0.001
TyG Index	12.12 ± 0.99	10.03 ± 0.63	<0.001
Vascular function			
CAVI	8.79 ± 1.28	8.72 ± 1.29	0.290
CAVI ≥ 9	443 (45.7)	420 (43.2)	0.267
baPWV, m/s	14.71 ± 2.62	14.29 ± 2.67	<0.001
baPWV ≥ 14.5, m/s	459 (47.3)	389 (40.0)	0.001

Continuous variables are displayed as means ± SD. Categorical variables are shown as number and %. MD: Mediterranean diet; SBP: systolic blood pressure; DBP: diastolic blood pressure; LDL: low-density lipoprotein; HDL: high-density lipoprotein; FPG: fasting plasma glucose; HbA1c: glycosylated hemoglobin; BMI: body mass index; WC: waist circumference; TGC: triglycerides. TyG Index: triglycerides/glucose index; CAVI: cardio–ankle vascular index; baPWV: brachial–ankle pulse wave velocity. *p* value: differences between subjects with and without insulin resistance.

**Table 4 nutrients-16-03106-t004:** General characteristics of female patients with and without IR.

	Insulin Resistance (n = 728)	No Insulin Resistance (n = 730)	*p* Value
Mediterranean diet			
MD, total score	5.72 ± 1.86	6.36 ± 2.05	<0.001
MD adherence, n (%)	241 (33.1)	346 (47.4)	<0.001
Conventional risk factors			
Age, years	62.47 ± 7.64	57.60 ± 11.43	<0.001
SBP, mmHg	133.87 ± 18.06	123.56 ± 21.84	<0.001
DBP, mmHg	81.93 ± 10.26	77.24 ± 10.85	<0.001
Hypertension, n (%)	532 (73.1)	312 (42.7)	<0.001
Antihypertensive drugs, n (%)	422 (58.0)	227 (31.1)	<0.001
Total cholesterol, mg/dL	228.83 ± 43.46	211.34 ± 40.29	<0.001
LDL cholesterol, mg/dL	139.73 ± 37.72	124.66 ± 32.45	<0.001
HDL cholesterol, mg/dL	50.33 ± 10.98	64.22 ± 16.67	<0.001
Triglycerides, mg/dL	164.04 ± 63.08	79.25 ± 22.39	<0.001
Dyslipidemia, n (%)	664 (91.2)	400 (54.8)	<0.001
Lipid–lowering drugs, n (%)	276 (37.9)	150 (20.5)	<0.001
FPG, mg/dL	115.13 ± 38.94	85.27 ± 11.53	<0.001
HbA1c, %	6.38 ± 1.29	5.48 ± 0.47	<0.001
Diabetes mellitus, n (%)	268 (36.8)	37 (5.1)	<0.001
Hypoglycemic drugs, n (%)	203 (27.9)	31 (4.2)	<0.001
Weight, kg	74.18 ± 13.54	66.19 ± 11.85	<0.001
Height, cm	156.38 ± 6.64	157.82 ± 6.66	<0.001
BMI, kg/m^2^	30.33 ± 5.18	26.62 ± 4.74	<0.001
WC, cm	99.56 ± 11.81	89.07 ± 11.79	<0.001
Obesity, n (%)	344 (47.3)	148 (20.3)	<0.001
TyG Index	11.85 ± 1.05	9.61 ± 0.70	<0.001
Vascular function			
CAVI	8.65 ± 1.20	8.09 ± 1.25	<0.001
CAVI ≥ 9	295 (40.5)	1667 (22.9)	<0.001
baPWV, m/s	15.18 ± 2.79	13.32 ± 2.77	<0.001
baPWV ≥ 14.5, m/s	401 (55.1)	226 (31.0)	<0.001

Continuous variables are displayed as means ± SD. Categorical variables are shown as number and %. MD: Mediterranean diet; SBP: systolic blood pressure; DBP: diastolic blood pressure; LDL: low-density lipoprotein; HDL: high-density lipoprotein; FPG: fasting plasma glucose; HbA1c: glycosylated hemoglobin; BMI: body mass index; WC: waist circumference; TGC: triglycerides; TyG Index: triglycerides/glucose index; CAVI: cardio–ankle vascular index; baPWV: brachial–ankle pulse wave velocity. *p* value: differences between subjects with and without insulin resistance.

**Table 5 nutrients-16-03106-t005:** Correlation between the Mediterranean diet and cardiovascular risk factors and arterial function parameters, overall, in subjects with and without insulin resistance, and by gender.

MD (Total Score)	Overall (n =)	Women (n =)	Men (n =)
Overall			
Age, years	0.005	−0.025	0.029
SBP, mmHg	−0.123 **	−0.106 **	−0.108 **
DBP, mmHg	−0.172 **	−0.122 **	−0.185 **
Total cholesterol, mg/dL	−0.164 **	−0.174 **	−0.173 **
LDL cholesterol, mg/dL	−0.155 **	−0.161 **	−0.150 **
HDL cholesterol, mg/dL	0.170 **	0.174 **	0.132 **
Triglycerides, mg/dL	−0.165 **	−0.143 **	−0.164 **
FPG, mg/dL	−0.118 **	−0.139 **	−0.098 **
HbA1c, %	−0.118 **	−0.171 **	−0.077 **
BMI	−0.133 **	−0.153 **	−0.115 **
WC, cm	−0.173 **	−0.195 **	−0.116 **
CAVI	−0.082 **	−0.093 **	−0.053 *
baPWV, m/s	−0.102 **	−0.097 **	−0.100 **
With insulin resistance			
Age, years	0.120	0.111 *	0.103 **
SBP, mmHg	−0.053 *	−0.010	−0.063 *
DBP, mmHg	−0.149	−0.082 *	−0.171 **
Total cholesterol, mg/dL	−0.128 **	−0.134 **	−0.147 **
LDL cholesterol, mg/dL	−0.116 **	−0.112 **	−0.131
HDL cholesterol, mg/dL	0.096 **	0.081 *	0.067 *
Triglycerides, mg/dL	−0.100 **	−0.036	−0.110 **
FPG, mg/dL	−0.037	−0.071	−0.016
HbA1c, %	−0.052 *	−0.114 *	−0.017
BMI	−0.071 **	−0.133 **	−0.036
WC, cm	−0.113 **	−0.144 **	−0.057
CAVI	−0.000	0.007	0.004
baPWV, m/s	−0.004	0.015	−0.034
Without insulin resistance			
Age, years	−0.056 *	−0.043	−0.046
SBP, mmHg	−0.137 **	−0.114 **	−0.124 **
DBP, mmHg	−0.143 **	−0.097 **	−0.152 **
Total cholesterol, mg/dL	−0.153 **	−0.158 **	−0.158 **
LDL cholesterol, mg/dL	−0.164 **	−0.155 **	−0.157 **
HDL cholesterol, mg/dL	0.136 **	0.139 **	0.091 **
Triglycerides, mg/dL	−0.080 **	−0.105 **	−0.032
FPG, mg/dL	−0.130 **	−0.134 **	−0.107 **
HbA1c, %	−0.108 **	−0.168 **	−0.061
BMI	−0.111 **	−0.077 *	−0.126 **
WC, cm	−0.150 **	−0.144 **	−0.110 **
CAVI	−0.125 **	−0.118 **	−0.101 **
baPWV, m/s	−0.140 **	−0.108 **	−0.142 **

MD: Mediterranean diet; SBP: systolic blood pressure; DBP: diastolic blood pressure; LDL: low-density lipoprotein; HDL: high-density lipoprotein; FPG: fasting plasma glucose; HbA1c: glycosylated hemoglobin; BMI: body mass index; WC: waist circumference; TGC: triglycerides; CAVI: cardio–ankle vascular index; baPWV: brachial–ankle pulse wave velocity; TyG index: triglycerides/glucose index. Pearson coefficient. * *p* < 0.05; ** *p* < 0.01.

**Table 6 nutrients-16-03106-t006:** Association of the Mediterranean diet with the vascular function in participants with and without insulin resistance, overall and by gender. Multiple regression analysis.

All Subjects Included				
Overall	β	(95% CI)		*p*
BaPWV, m/s	−0.126	(−0.164 to −0.089)		<0.001
CAVI	−0.045	(−0.062 to −0.028)		<0.001
Men				
BaPWV, m/s	−0.147	(−0.196 to −0.098)		<0.001
CAVI	−0.046	(−0.067 to −0.024)		<0.001
Women				
BaPWV, m/s	−0.087	(−0.145 to −0.029)		0.004
CAVI	−0.044	(−0.070 to −0.018)		0.001
Without insulin resistance				
Overall				
BaPWV, m/s	−0.128	(−0.177 to −0.079)		<0.001
CAVI	−0.050	(−0.072 to −0.028)		<0.001
Men				
BaPWV, m/s	−0.154	(−0.222 to −0.086)		<0.001
CAVI	−0.046	(−0.076 to −0.016)		0.002
Women				
BaPWV, m/s	−0.093	(−0.163 to −0.023)		0.009
CAVI	−0.053	(−0.086 to −0.020)		0.002
With insulin resistance				
Overall				
BaPWV, m/s	−0.080	(−0.139 to −0.022)		0.007
CAVI	−0.035	(−0.061 to −0.009)		0.007
Men				
BaPWV, m/s	−0.111	(−0.185 to −0.038)		0.003
CAVI	−0.042	(−0.074 to −0.009)		0.012
Women				
BaPWV, m/s	−0.030	(−0.126 to 0.067)		0.549
CAVI	−0.024	(−0.066 to 0.018)		0.265

Multiple regression. Dependent variables: baPWV m/s and CAVI. Independent variable: MD. Adjustment variables: age, sex, and hypotensive, hypoglycemic, and lipid-lowering drugs. MD: Mediterranean diet; baPWV: brachial–ankle pulse wave velocity; CAVI: cardio–ankle vascular index.

## Data Availability

The data supporting the findings of this study are available on ZENODO under the DOI https://doi.org/10.5281/zenodo.12166167 (accessed on 24 June 2024).

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
