# Peer review of "Relationship between the Mediterranean Diet and Vascular Function in Subjects with and without Increased Insulin Resistance"

_nutrients, 2024, doi:10.3390/nu16183106_

Round 1

Reviewer 1 Report

Comments and Suggestions for Authors

Very nice study, well written and presented. I just want to ask the authors to briefly clarify the nature of the EVA, MARK and EVIDENT studies. This would help the readers to better identify the target populations included in these studies. 

Author Response

Yes

Reviewer 2 Report

Comments and Suggestions for Authors

Authros examined the effects of a Mediterranean diet on vascular function in both insulin-resistant and non-insulin-resistant patients. 

It has been demonstrated that the Mediterranean diet could improve cardiovascular function, especially by improving endothelial health. This diet plan is linked to a lower risk of cardiovascular disease. It has been demonstrated that the Mediterranean diet has a major impact on insulin resistance, especially in those with metabolic diseases and obesity. The Mediterranean diet is a helpful dietary strategy for controlling and preventing type 2 diabetes, according to research, since it is associated with increased insulin sensitivity and decreased insulin resistance. The findings of this study indicate that, with the exception of the group of women who had insulin resistance, all of the groups evaluated showed a negative correlation between the arterial stiffness parameters and adherence to the Mediterranean diet.

The methodology is quite complex, and the authors should specify if it is a secondary study, a retrospective study, and so on. It is not common to find data from 3 different datasets. Variables from 3 RCTs on different populations are considered; different operators collected and elaborated those data. I suggest considering this topic in the limitations.

Moreover, I recommended to writers that they take into account the connection between insulin resistance and depression in their writing. In clinical and research contexts, its recognition is growing. Research points to the possibility that insulin resistance influences depression severity and response to treatment, in addition to being a side effect of depression (10.1007/s43440-024-00621-5); it is a crucial point, especially in older adults (10.1186/s12877-020-01730-5).

I appreciate the availability of Dataset on Zenodo. 

Author Response

Yes

Reviewer 3 Report

Comments and Suggestions for Authors

This is a very interesting study investigating the effects of mediterranean diet with vascular stiffness in women and men. The authors are commended on the study design, the pooling of the different study populations and the hypothesis that could potentially explain the cardio-protection of this way of eating. Please find below my specific comments:

-Regarding insulin resistance, what was the rationale for classifying IR as TyGlu index greater than median? This becomes even more important in the setting of your population that was largely devoid of cardiovascular disease and therefore using TyG values in these individuals may over-represent the presence of insulin resistance.

-Line 190: do you mean anti-diabetes drugs?

-Line 201: please state this sentence in English.

-You mention for insulin resistance that the median was considered a cut-off point, yet you reported all group variables as mean rather than median. Was the distribution of all variables  normal and how was normality assessed (i.e. with which test)?

-Could you please provide a legend for Fig.3? What do red vs blue vs pink bars represent?

-Please be consistent in using the abbreviation MD for mediterranean diet; the discussion section contain quite a few uses of DM rather than MD

-It is intriguing to consider the use of TyG ratio in primary care as a tool to identify individuals with insulin resistance, however it is difficult to envision how this could be implemented given that there isn't a specific cut-off value that the primary care physicians can use as a benchmark. Could you please elaborate as to some of your suggestions of how that could be overcome? 

Comments on the Quality of English Language

Minor English language corrections necessary.

Author Response

Yes

Round 2

Reviewer 2 Report

Comments and Suggestions for Authors

The authors addressed all suggestions; the manuscript is suitable for publication.